# Integrating Bioinformatics and Experimental Validation Identifies SCD as a Ferroptosis-Related Immune Regulator and Therapeutic Target in Crohn’s Disease

**DOI:** 10.3390/ijms27010019

**Published:** 2025-12-19

**Authors:** Jingtong Wu, Lixiang Zhou, Dongmei Qiu, Tao Wei, Chaohui Xie, Ermei Chen, Mengjie Lin, Yanyun Fan

**Affiliations:** 1Department of Gastroenterology, Clinical Research Center for Gut Microbiota and Digestive Diseases of Fujian Province, The National Key Clinical Specialty, Zhongshan Hospital of Xiamen University, School of Medicine, Xiamen University, Xiamen 361004, China; jingtongw@foxmail.com (J.W.); zhoulixiang8@163.com (L.Z.); 24520240157796@stu.xmu.edu.cn (D.Q.); xmxzhh@126.com (C.X.); cemsmiles@163.com (E.C.); 2Xiamen Key Laboratory of Intestinal Microbiome and Human Health, Zhongshan Hospital of Xiamen University, Xiamen 361004, China; 3Department of Digestive Disease, Institute for Microbial Ecology, School of Medicine, Xiamen University, Xiamen 361004, China; 21620231153609@stu.xmu.edu.cn; 4Department of Pathology, Zhongshan Hospital of Xiamen University, School of Medicine, Xiamen University, Xiamen 361004, China

**Keywords:** Crohn’s disease (CD), differentially expressed genes (DEGs), ferroptosis, ferroptosis-related DEGs (FEDGs), immune infiltration, Stearoyl coenzyme A desaturase

## Abstract

This study investigates the role of ferroptosis-related genes (FRGs) in the intestinal inflammation of Crohn’s disease (CD). Through integrated bioinformatics and experimental validation, we identified differentially expressed genes from RNA-seq data and intersected them with known FRGs to obtain ferroptosis-related differentially expressed genes (FEDGs). Functional enrichment and immune infiltration analyses were performed, and seven hub FEDGs were selected using machine learning. A diagnostic model based on these genes showed strong predictive ability. Immune analysis revealed significant associations with macrophages, neutrophils, dendritic cells, and CD4+ T cells. Protein expression of key hub genes was validated in clinical CD samples and a DSS-induced colitis model. Importantly, localized inhibition of SCD alleviated disease severity in experimental colitis. These findings highlight the involvement of ferroptosis in CD immune dysregulation and propose SCD as a potential therapeutic target.

## 1. Introduction

Crohn’s disease (CD) is an inflammatory bowel disease (IBD), a chronic autoimmune disorder characterized by dysregulated innate and adaptive immune responses [1,2]. Clinical manifestations of CD include abdominal pain, chronic diarrhea, fistula, abscess, anal lesions, and systemic symptoms, all of which significantly impair patients’ quality of life [3,4,5,6]. Over the past few decades, the global incidence of CD has risen, particularly in developing countries [7,8]. The lack of specific early clinical features, coupled with the limited efficacy of current therapies, complicates early diagnosis and frequently leads to severe complications [9,10]. Therefore, the identification of reliable biomarkers is urgently needed to improve early detection and deepen our understanding of CD pathogenesis, ultimately facilitating better clinical management.

Ferroptosis is a form of regulated cell death (RCD) driven by iron-dependent accumulation of lethal lipid peroxides [11,12]. This nonapoptotic type of RCD has been implicated in various pathological conditions, such as cancer, stroke, intracerebral hemorrhage, traumatic brain injury, kidney degeneration, neurodegenerative illnesses (including Huntington’s, Alzheimer’s, and Parkinson’s diseases), and ischemia–reperfusion damage [12,13,14,15,16,17,18]. Recently, ferroptosis has attracted growing attention in IBD research, with accumulating evidence supporting its role in the development of CD [19,20,21,22]. However, the precise mechanisms through which ferroptosis contributes to CD pathology remain incompletely understood. Targeting ferroptosis may represent a promising therapeutic strategy, underscoring the need to investigate ferroptosis-related genes (FRGs) involved in intestinal inflammation.

In this study, we aimed to identify and validate ferroptosis-associated gene signatures in CD using integrated bioinformatics approaches and experimental validation. We retrieved RNA-seq datasets from the Gene Expression Omnibus (GEO) database, comprising samples from both healthy individuals and CD patients. We identified differentially expressed genes (DEGs) and intersected them with a curated list of FRGs to obtain ferroptosis-related differentially expressed genes (FEDGs). Functional enrichment analyses, including Gene Ontology (GO) and Kyoto Encyclopedia of Genes and Genomes (KEGG) pathway analyses, were performed to elucidate the biological roles of these genes.

Using CIBERSORT and ssGSEA, we evaluated immune cell infiltration in intestinal mucosa and examined its correlation with FEDG expression. Key FEDGs were selected via LASSO regression, and a diagnostic model was established based on ROC analysis. Finally, we validated protein expression of hub genes using immunohistochemistry (IHC) in clinical samples and a dextran sulfate sodium (DSS)-induced murine colitis model, and a 2,4,6-trinitrobenzenesulfonic acid (TNBS)-induced colitis model. Additionally, we assessed the therapeutic potential of inhibiting a central hub gene, SCD, in vivo.

Our comprehensive approach provides new insights into the interplay between ferroptosis and immune dysregulation in CD, highlighting promising diagnostic markers and therapeutic targets.

## 2. Results

### 2.1. Identification of DEGs Between CD Patients and Healthy Subjects

Two datasets (GSE75214 and GSE112366) containing gene expression profiles in intestinal mucosal samples from both healthy subjects (48 in total) and CD patients (437 in total) were retrieved. The “PreprocessCore” package in R was utilized to normalize the RNA-seq data. Based on the normalized data, we identified 3906 upregulated and 6137 downregulated DEGs in GSE75214, and 2500 upregulated and 2861 downregulated DEGs in GSE112366. The gene expression patterns of these DEGs are illustrated in Figure 1A,C. The top 20 most significant DEGs for each dataset are presented as heatmaps in Figure 1B,D, including *MMP1*, *MMP3*, *S100A8*, *CXCL5*, *CXCL13*, *CDHR1*, and *PAQR5*.

### 2.2. Characterization of the Immune Landscape in the Intestinal Mucosal Tissues

A Venn diagram revealed 1768 DEGs common to both datasets, of which 911 were upregulated and 857 were downregulated (Figure 2A,B). Functional enrichment analyses showed that these common DEGs were primarily associated with biological processes (BP) such as “neutrophil activation” and “neutrophil-mediated immunity”. Cellular component (CC) terms included “collagen-containing extracellular matrix” and “vesicle lumen”, while molecular function (MF) terms were enriched in “cytokine receptor binding” and “extracellular matrix structural constituent” (Figure 2C). KEGG pathway analysis indicated significant enrichment in “cytokine-cytokine receptor interaction”, “chemokine signaling pathway”, “TNF signaling pathway”, “IL-17 signaling pathway”, and “B cell receptor signaling pathway” (Figure 2D).

### 2.3. Identification and Functional Annotation of Ferroptosis-Related DEGs

We intersected the common DEGs with 388 ferroptosis-related genes from the FerrDb database, identifying 59 ferroptosis-related DEGs (FEDGs) (Figure 3A and Appendix A). *T*-test analysis revealed that 8 of these FEDGs did not show significant expression differences between groups (Appendix A), though they were retained for subsequent analysis. GO enrichment analysis indicated that FEDGs were enriched in BP terms such as “response to oxidative stress” and “neuron death”, CC terms including “peroxisome” and “NADPH oxidase complex”, and MF terms like “ubiquitin protein ligase binding” and “peroxidase activity” (Figure 3B). KEGG analysis revealed enrichment in pathways such as “lipid and atherosclerosis”, “mTOR signaling pathway”, “ferroptosis”, “TNF signaling pathway”, and “Th17 cell differentiation” (Figure 3C). Notably, 28 FEDGs were involved in the most significantly enriched pathways, with seven (*ATG7*, *GPX4*, *SLC7A11*, *SLC3A2*, *MAP1LC3A*, *ACSL4*, and *SAT1*) directly participating in ferroptosis (Figure 3D).

### 2.4. Identification of Hub FEDGs

To identify the hub genes, logistic LASSO regression was first performed on the FEDGs in the GSE75214 and GSE112366 datasets separately. This analysis identified 17 and 22 hub FEDGs in the two datasets, respectively (Figure 4A,B). Among them, 7 hub FEDGs, namely *SCD*, *ATF4*, *SAT1*, *RPL8*, *MUC1*, *MTDH*, and *ATP6V1G2*, were shared by both datasets (Figure 4C). We next examined the interactions and correlations between these 7 hub FEDGs. A circular chord plot revealed a complex co-expression network among these genes, with MUC1 exhibiting particularly wide connecting arcs, suggesting its central role and high expression level within the network (Figure 5A). Pearson correlation analysis further confirmed strong and significant correlations between their expression levels. Specifically, the expression of *SAT1* and *RPL8*, as well as *MUC1* and *ATP6V1G2*, showed strong negative correlations in both datasets. Conversely, the expression level of *MUC1* showed a strong positive correlation with that of *SCD*. The correlation between *MTDH* and *ATF4* was also positive (Figure 5B). Subsequently, ROC curves were generated for these hub FEDGs to evaluate their diagnostic potential for CD, and the models demonstrated high predictive accuracy (Figure 5C,D).

### 2.5. Associations Between the Hub FEDGs and Immune Cell Infiltration and Pathways

We employed CIBERSORT to deconvolute the immune cell infiltration landscape in the intestinal mucosal tissues of healthy subjects and CD patients from the GSE75214 dataset (Appendix A). The analysis revealed that T cells, monocytes, macrophages, neutrophils, natural killer (NK) cells, dendritic cells, and mast cells were the major infiltrating immune cell types. A more detailed correlation analysis, plotting the correlation coefficients and significance (*p*-values) for each hub FEDG against specific immune cell types, further confirmed that most hub FEDGs were significantly associated with the infiltration of multiple immune cells. For instance, *SCD* expression showed a strong positive correlation with the infiltration of M1 macrophages (Figure 6).

These results collectively indicate that the infiltration of specific immune cells and the expression of hub FEDGs are intricately linked, contributing to an altered immunological microenvironment in the intestinal mucosa of CD patients.

### 2.6. Functional Enrichment Analysis of the Hub FEDGs in CD

To elucidate the biological pathways influenced by the hub FEDGs in CD pathogenesis, we first analyzed the correlation between each hub FEDG and all other genes in the GSE75214 dataset (Appendix A). This was followed by a Reactome-based Gene Set Enrichment Analysis (GSEA) using the single-gene correlation results.

The GSEA revealed that *MTDH* was prominently involved in numerous immune-related pathways, including the adaptive and innate immune systems, cytokine signaling pathways, and neutrophil degranulation, highlighting its potential multifaceted role in immune regulation (Figure 7). Functional enrichment analysis for all seven hub FEDGs uncovered their diverse involvement in key biological processes. *ATF4* was associated with extracellular matrix organization and RNA metabolism; *ATP6V1G2* with the neuronal system; *MTDH* with cell cycle regulation; *MUC1* with vitamin and cofactor metabolism; *RPL8* with translational processes; *SAT1* with protein metabolism; and *SCD* with cell cycle regulation (Figure 7). These findings suggest that the hub FEDGs contribute to CD pathogenesis through regulating not only ferroptosis and immunity but also a wider array of cellular functions.

### 2.7. Determination of the Protein Expression Levels in Intestinal Mucosal Tissues

To validate our bioinformatic findings at the protein level, we performed immunohistochemical (IHC) analysis on intestinal mucosal tissues from Crohn’s disease (CD) patients and healthy controls (Figure 8). The results demonstrated that, compared to healthy tissues, the protein level of GPX4, a key inhibitor of ferroptosis, was markedly reduced in the CD group. In contrast, the expression of SCD, ATF4, MUC1, and RPL8 were significantly elevated in CD tissues. No significant differences were observed in the protein levels of SAT1, MTDH, or ATP6V1G2 between the two groups. Quantitative analysis of the IHC staining (positive area percentage) confirmed these observations with high statistical significance (Figure 8). Notably, the protein-level validation for SCD and MUC1 was highly consistent with their mRNA expression patterns observed in the transcriptomic datasets, reinforcing their roles as central hub genes in CD pathogenesis.

To further evaluate the relevance of hub FEDGs identified from human transcriptomes, we examined their protein expression in intestinal tissues from a dextran sulfate sodium (DSS)-induced murine model of Crohn’s disease and healthy controls using immunohistochemistry (IHC). As illustrated in Appendix A, quantitative IHC revealed distinct expression patterns. Consistent with enhanced ferroptosis in colitis, GPX4 expression was significantly decreased in DSS-treated mice compared with controls, supporting the activation of ferroptosis in this model. In contrast to its upregulation in human CD samples, MUC1 was notably downregulated in the DSS model, suggesting limited translational relevance of MUC1 in this specific murine system and highlighting potential interspecies or model-specific disparities. Notably, SCD exhibited no significant difference between groups, indicating stable protein expression under inflammatory conditions.

Given its conserved expression across species and absence of conflict with human data, SCD was selected for further functional exploration in subsequent animal experiments to clarify its role in ferroptosis and intestinal inflammation.

### 2.8. Therapeutic Effect of Local SCD Inhibition in DSS-Induced and TNBS-Induced Murine Model of Colitis

To functionally validate the role of SCD in intestinal inflammation and its potential interaction with ferroptosis, we employed both dextran sulfate sodium (DSS)-induced and 2,4,6-trinitrobenzenesulfonic acid (TNBS)-induced murine colitis models. Given our prior bioinformatic and immunohistochemical findings that highlighted SCD as a central hub gene in CD pathogenesis, and considering previous reports that hepatic SCD can influence intestinal inflammation, we sought to target SCD specifically within the colon to avoid systemic effects. To this end, we administered a specific SCD inhibitor via intracolonal injection (i.c.) throughout the DSS and TNBS treatment period.

In the DSS-induced colitis model, local inhibition of SCD significantly ameliorated the severity of colitis. The SCD inhibitor group demonstrated a markedly attenuated loss of body weight (Figure 9A), preserved colon length (Figure 9B), and a significant reduction in spleen weight, a hallmark of systemic inflammation (Figure 9C). Furthermore, histopathological assessment revealed a substantial improvement in mucosal architecture, with reduced epithelial damage, crypt loss, and inflammatory cell infiltration in the inhibitor-treated group, which was corroborated by a significantly lower histopathological score (Figure 9D).

We therefore employed the TNBS-induced colitis model to further validate the therapeutic potential of SCD inhibition, as this model better replicates the transmural inflammation characteristic of CD. Consistent with the findings from the DSS model, local SCD inhibition in the TNBS model profoundly alleviated disease severity. This was evidenced by a significantly lower Disease Activity Index (DAI) (Figure 10B), preserved colon length (Figure 10C,D), and a notable improvement in histopathological scores (Figure 10E,F).

To investigate the immunomodulatory mechanism underlying this protective effect, we analyzed the colonic expression profile of key inflammatory mediators. Local SCD inhibition potently suppressed the colonic expression of a broad spectrum of pro-inflammatory cytokines and chemokines in both models. Treatment with the SCD inhibitor dramatically suppressed the expression of critical pro-inflammatory cytokines (*Tnf*, *Ifng*, *Il1b*, *Il17a*), chemokines (*Cxcl1*, *Cxcl2*, *Cxcl3*, *Cxcl5*, *Cxcl10*) (Figure 9E–N for DSS; Figure 10G–P for TNBS). Conversely, the expression of the anti-inflammatory cytokine *Il10* was upregulated upon SCD inhibition (Figure 9I for DSS; Figure 10K for TNBS), indicating a shift in the immune balance towards an anti-inflammatory state.

We exam the protein expression of key hub genes in both the DSS-induced and TNBS-induced colitis models following treatment. A consistent pattern was observed across both models: the protein level of GPX4, a master inhibitor of ferroptosis, was significantly restored in the SCD inhibitor-treated groups compared to the colitis control groups (Appendix A). In contrast, the protein expression of SCD itself, MUC1, ATP6V1G2, and SAT1, remained unchanged upon treatment. These findings strongly suggest that the therapeutic effect of SCD inhibition is mediated by the specific rescue of GPX4 protein expression, which in turn counteracts the ferroptosis process in intestinal inflammation.

These results collectively demonstrate that local inhibition of SCD in the colon is sufficient to confer a strong protective effect against DSS-induced colitis, alleviating both pathological hallmarks and the associated dysregulation of the inflammatory response. The consistent efficacy observed in two distinct models suggests that SCD activity within the colonic mucosa itself plays a pivotal pro-inflammatory role during intestinal inflammation, primarily by modulating the expression of cytokines and chemokines, making it a compelling therapeutic target for Crohn’s disease.

## 3. Discussion

By integrating multi-omics approaches, this study reveals a crucial connection between ferroptosis and immune microenvironment dysregulation in Crohn’s disease (CD), identifying seven hub ferroptosis-related differentially expressed genes (FEDGs): SCD, ATF4, SAT1, RPL8, MUC1, MTDH, and ATP6V1G2. Functional validation confirmed that inhibiting SCD ameliorated disease severity and suppressed pro-inflammatory signaling in experimental colitis, highlighting its therapeutic potential.

Enrichment analysis indicated that both commonly differentially expressed genes (DEGs) and FEDGs were significantly associated with immune pathways (e.g., TNF and IL-17 signaling) and ferroptosis-related processes (e.g., response to oxidative stress). The overlap between ferroptosis and immune inflammation in CD suggests a potential vicious cycle wherein ferroptosis-derived DAMPs or lipid peroxidation products amplify pro-inflammatory signaling.

Network and correlation analyses indicated that these hub FEDGs function cooperatively within a shared regulatory network. MUC1 and SCD exhibited strong co-expression, suggesting the possibility of collaborative function in pro-inflammatory or metabolic pathways. It is noteworthy that SCD expression exhibited a robust correlation with M1 macrophage infiltration, while MTDH demonstrated a connection with a more extensive array of immune pathways, including cytokine signaling and neutrophil activation. SAT1 and RPL8 exhibited a robust negative correlation, indicating a potential compensatory regulatory mechanism between the stress response and translational control. Notably, the unique configuration of the diiron center in SCD and its susceptibility to iron loss during catalysis could represent a novel mechanism linking its metabolic function to ferroptosis and inflammatory amplification [23].

An intriguing aspect of our study lies in the divergent protein expression patterns of hub genes between human CD samples and the DSS-induced murine colitis model. Although MUC1 exhibited significant upregulation in human diseased tissues, it demonstrated downregulation in the murine model. Concurrently, SCD protein levels remained stable in mice despite being elevated in human CD. The observed species-specific and model-dependent differences may arise from distinct pathophysiological contexts, the chronic, complex immune milieu of human CD versus the acute, chemical injury-driven DSS model or from post-transcriptional regulation that decouples mRNA abundance from protein translation. This observation is further supported by previous studies showing that the inflammatory consequences of SCD modulation can be highly environment-dependent [24,25], and highlights the importance of considering species-specific differences when translating findings from murine models to human disease.

Our functional validation experiments demonstrate that local inhibition of SCD profoundly reshapes the cytokine/chemokine network in colitic mice, corroborating our earlier bioinformatic findings. Specifically, suppression of SCD led to broad downregulation of key pro-inflammatory cytokines (e.g., *Tnf*, *Il1b*, *Il17a*) and chemokines (e.g., *Cxcl1*, *Cxcl5*, *Cxcl10*), while enhancing the expression of the anti-inflammatory cytokine *Il10*. This immunomodulatory shift corresponds closely with the “cytokine-cytokine receptor interaction” pathway, one of the most significantly enriched KEGG pathways identified in our analysis of common DEGs. Moreover, the attenuated infiltration of immune cells such as neutrophils and M1 macrophages upon SCD inhibition resonates with the reduced activity of related signaling pathways (e.g., IL-17 and chemokine signaling), thereby functionally validating the centrality of SCD in modulating immune responses in colitis. Collectively, these results link SCD-driven ferroptosis to the dysregulated immune microenvironment via cytokine/chemokine networks and highlight SCD as a potential therapeutic target for reversing immune imbalance in CD.

Despite these insights, our study has limitations that should be noted. First, the causal relationships among the seven hub FEDGs remain unclear. Experimental validation—such as knockdown or overexpression studies—are needed to confirm regulatory interactions, for instance between MUC1 and SCD. Second, although SCD inhibition alleviated colitis, its specific effects on ferroptosis markers (e.g., GSH, MDA, lipid ROS) were not quantitatively measured. Future work should directly evaluate these indicators to clarify whether SCD promotes inflammation primarily via ferroptosis or other mechanisms. Third, the role of MUC1 remains ambiguous due to its inconsistent expression across species. Investigations using cell-specific knockouts or human organoids could help clarify its context-dependent functions. Finally, the acute DSS model may not fully reflect the chronicity and heterogeneity of human Crohn’s disease. Validation in spontaneous chronic models or patient-derived cells would strengthen clinical relevance.

In summary, our study significantly advances the understanding of Crohn’s disease pathogenesis by systematically integrating multi-omics data with functional validation to delineate the role of ferroptosis-related genes, particularly SCD, in modulating intestinal immunity and inflammation. We identified and characterized seven hub FEDGs, revealing their cooperative involvement not only in ferroptosis but also in immune signaling and cellular stress responses. We demonstrated that local inhibition of SCD effectively attenuates colitis severity and rebalances the cytokine-chemokine network in vivo, underscoring its role in disease progression. These findings may establish a mechanistic link between ferroptosis and chronic inflammation, underscoring SCD as a promising therapeutic target. The translational potential of targeting SCD is further supported by its conserved dysregulation in human disease, offering a novel strategy for immunomodulatory intervention in Crohn’s disease and potentially other ferroptosis-associated inflammatory disorders.

## 4. Materials and Methods

### 4.1. Data Collection

Two CD-related datasets, GSE112366 and GSE75214, were obtained from the GEO database, while the ferroptosis-related genes (FRGs) were obtained from the FreeDb online database (http://www.datjar.com:40013/bt2104/, access date: 1 June 2022).

### 4.2. Identification of Differentially Expressed Genes (DEGs)

Raw data from both datasets were normalized using the “PreprocessCore” package in R (version 4.2.0) to eliminate technical biases. Subsequently, the “limma” package was employed to identify DEGs between CD patients and healthy controls for each dataset individually [26]. Genes with an adjusted *p*-value < 0.05 (Benjamini & Hochberg method) and an absolute log2 fold change (|log_2_FC|) > 0 were considered statistically significant. The shared DEGs common to both datasets were visualized using a Venn diagram. These common DEGs were then intersected with the FRGs list to identify Ferroptosis-related DEGs (FEDGs), which was also illustrated with a Venn diagram.

### 4.3. Functional Enrichment Analysis

GO functional annotation and KEGG pathway enrichment analysis of the DEGs and FEDGs were performed by using the “clusterProfiler” package in R [27]. GO terms were categorized into Biological Processes (BP), Cellular Components (CC), and Molecular Functions (MF). A significance threshold of an adjusted *p*-value < 0.05 was set for all enrichment analyses [28].

### 4.4. Screening and Validation of Hub FEDGs

The Least Absolute Shrinkage and Selection Operator (LASSO) logistic regression algorithm, implemented via the “glmnet” package in R [29], was applied to the normalized expression matrix of the FEDGs to identify hub genes with the strongest diagnostic potential. This analysis was performed separately on each dataset (GSE75214 and GSE112366). The intersection of hub FEDGs from both datasets was taken to identify the final robust hub genes. The diagnostic accuracy of each hub FEDG was evaluated by constructing Receiver Operating Characteristic (ROC) curves and calculating the Area Under the Curve (AUC) using the “pROC” package.

### 4.5. Immune Cell Infiltration Analysis

The CIBERSORT package [30] was utilized to ascertain the distinctions in the infiltration of 22 subtypes of immune cells between CD patients and healthy individuals in the GSE75214 dataset, with the objective of investigating the association between the expression of the hub genes and the immunological landscape in intestinal mucosa. Subsequently, the ssGSEA method was employed to analyze 22 immune gene sets corresponding to different types of immune cells using the R package “gsva” in order to confirm the immunological characteristics of CD [31,32]. The association between the expression of each hub FEDG and the fraction of different types of immune cells was also examined with the “clusterProfiler” package in R (*p* < 0.05).

### 4.6. IHC Staining Analysis of Formalin-Fixed Intestinal Mucosal Samples from CD Patients and Healthy Subjects

Formalin-fixed, paraffin-embedded intestinal mucosal tissue sections from 40 CD patients and 10 healthy controls were obtained from Zhongshan Hospital of Xiamen University. IHC staining was performed to detect the protein expression levels of the hub FEDGs (GPX4, MUC1, SCD, MTDH, ATF4, SAT1, RPL8, ATP6V1G2) following standard protocols. The primary antibodies used were: rabbit anti-GPX4 (1:200; Absin, Shanghai, China), mouse anti-MUC1 (1:200; Absin), rabbit anti-SCD (1:200; Absin), rabbit anti-MTDH (1:300; Absin, Shanghai, China), rabbit anti-ATF4 (1:200; Proteintech, Wuhan, China), rabbit anti-SAT1 (1:200; Bioss, Woburn, MA, USA), rabbit anti-RPL8 (1:200; Proteintech, Wuhan, China), and rabbit anti-ATP6V1G2 (1:200; Bioss, Woburn, MA, USA). For quantitative analysis, five random fields per section were captured. The staining intensity and integrated density (IntDen) were analyzed using the ImageJ software (version 1.54) with the IHC Profiler plugin. The percentage of positive staining area was calculated for statistical comparison between groups.

### 4.7. Animal Model and Therapeutic Intervention

Male C57BL/6 mice (6–8 weeks old) were obtained from the Animal Center of Xiamen University. The mice were housed under specific pathogen-free conditions with a 12/12 h light/dark cycle and had free access to food and water. All experimental procedures were conducted in accordance with guidelines approved by the Animal Ethics Committee of Xiamen University.

The DSS-induced colitis model was established as described in our previous study [33]. An experimental colitis model was established in C57BL/6 mice by administering 3% (*w*/*v*) dextran sulfate sodium (DSS, 60316, Yeasen, Shanghai, China) in drinking water for 5 days. To investigate the role of SCD, a hub FEDG, mice with DSS-induced colitis received intracolonal injections of a specific SCD inhibitor (HY-155033, 1 mg/kg in PBS with 0.1% DMSO) or vehicle control daily throughout the DSS treatment period.

For the TNBS-induced colitis model, the protocol was adapted from established methods [34,35]. Mice were randomly divided into three groups: (1) Blank group (2) Ctrl group (3) SCD-inhibitor group. On day 0, mice in the latter two groups were pre-sensitized, and colitis was subsequently induced with 2.5% TNBS (MB5523, Meilunbio, Dalian, China) in 50% ethanol by rectal injection 7 days after presensitization. Starting from day 1, mice in the SCD-inhibitor4 group received daily intracolonal injections of the SCD inhibitor (1 mg/kg) for preventive purposes, while the Ctrl group received the vehicle. All mice were sacrificed on day 10.

Colon tissues were collected for histological scoring (H&E staining) and RNA extraction. Total RNA was extracted using TRIzol reagent (Thermo Fisher Scientific, Waltham, MA, USA). Complementary DNA (cDNA) was synthesized using the cDNA Synthesis Kit (11141, Yeasen, Shanghai, China). Quantitative real-time PCR (qPCR) was then performed using qPCR SYBR Green Master Mix (11201, Yeasen, Shanghai, China). The expression levels of inflammatory cytokines (Tnf, Ifng, Il1b, Il6, Il17a, Il10) and chemokines (Cxcl1, Cxcl2, Cxcl3, Cxcl5, Cxcl10) were quantified by quantitative real-time PCR (qRT-PCR) and primer were listed in Appendix A.

### 4.8. Statistical Analysis

To compare gene expression levels between CD patients and healthy individuals, we employed Student’s *t*-test. To investigate the relationship between the expression levels of genes and the proportions of different types of infiltrating immune cells, we performed ssGSEA and Pearson correlation analysis. All statistical analyses were conducted using GraphPad Prism 9 and R software (Version 4.2.0). * *p* < 0.05, ** *p* < 0.01, and *** *p* < 0.001 represented three levels of statistical significance.

## 5. Conclusions

This study delineates the role of ferroptosis in Crohn’s disease by identifying seven hub ferroptosis-related differentially expressed genes, notably SCD, which are closely linked to immune dysregulation. We further demonstrate that local inhibition of SCD alleviates colitis severity and modulates inflammatory responses in vivo. These findings not only provide mechanistic insights into ferroptosis-associated intestinal inflammation but also highlight SCD as a promising diagnostic biomarker and therapeutic target for Crohn’s disease.

## Figures and Tables

**Figure 1 ijms-27-00019-f001:**
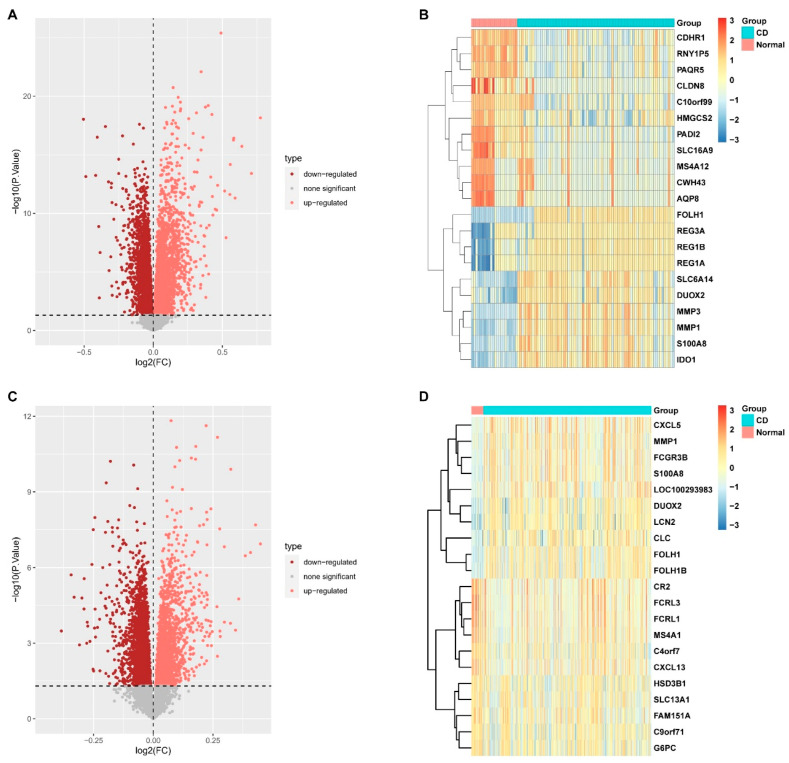
Identification of differentially expressed genes (DEGs) between healthy subjects and Crohn’s disease (CD) patients. (**A**) Volcano plot of DEGs from the GSE112366 dataset. (**B**) Heatmap displaying the expression patterns of the top 20 most significant DEGs from the GSE112366 dataset. (**C**) Volcano plot of DEGs from the GSE75214 dataset. (**D**) Heatmap of the top 20 most significant DEGs from the GSE75214 dataset. (DEGs were identified using the threshold *p* < 0.05 and |log_2_FC| > 0).

**Figure 2 ijms-27-00019-f002:**
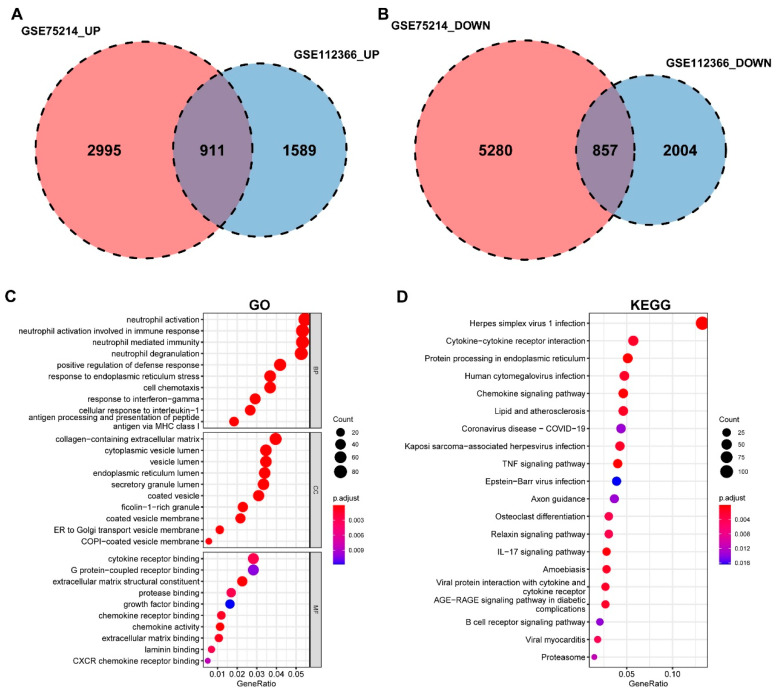
Functional enrichment analysis of differentially expressed genes (DEGs). (**A**) Venn diagram showing the common upregulated DEGs between the GSE75214 (2995 genes) and GSE112366 (1589 genes) datasets, with 911 overlapping genes. (**B**) Venn diagram showing the common downregulated DEGs between the GSE75214 (5280 genes) and GSE112366 (2004 genes) datasets, with 857 overlapping genes. (**C**) Bubble plot of significantly enriched Gene Ontology (GO) terms for the common DEGs, including molecular function (MF), cellular component (CC), and biological process (BP). (**D**) Bubble plot of the top enriched KEGG pathways.

**Figure 3 ijms-27-00019-f003:**
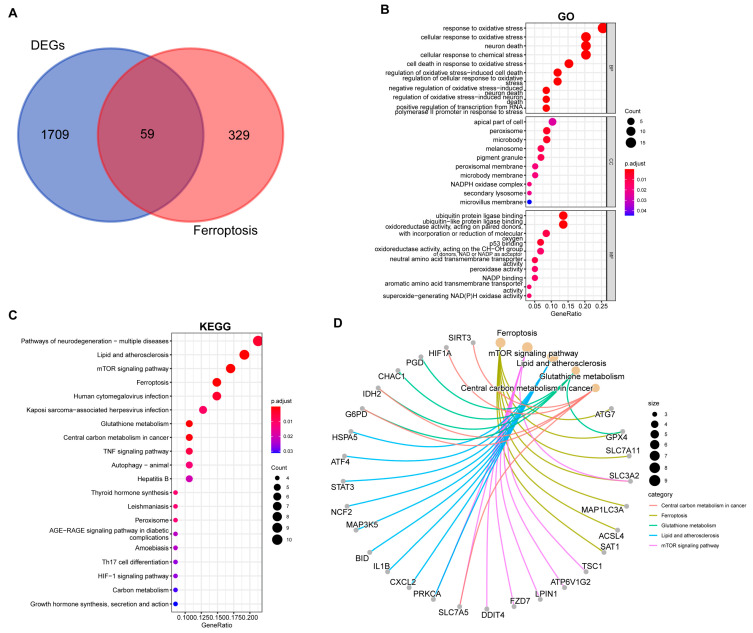
Functional enrichment analysis of ferroptosis-related differentially expressed genes (FEDGs). (**A**) Venn diagram illustrating the identification of FEDGs from the overlap between ferroptosis-related genes (FRGs) and common DEGs. Bubble plots display the significantly enriched Gene Ontology (GO) biological processes (**B**) and Kyoto Encyclopedia of Genes and Genomes (KEGG) pathways (**C**). (**D**) A network diagram visualizing the associations between the key enriched KEGG pathways and their corresponding FEDGs.

**Figure 4 ijms-27-00019-f004:**
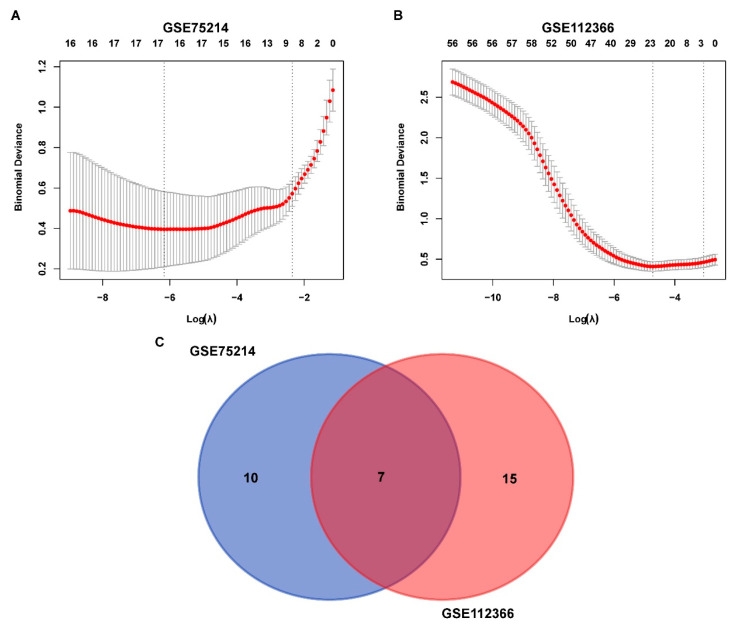
Identification and overlap of FEDGs in the GSE75214 and GSE112366 datasets. (**A**,**B**) Binomial deviance of genes plotted against the logarithm of lambda (λ) for the GSE75214 (**A**) and GSE112366 (**B**) datasets. The red line indicates the mean binomial deviance, and the gray area represents the deviance range. The two vertical dashed lines represent the optimal λ values selected by cross-validation, corresponding to the lambda.min (left) and the lambda.1se (right). (**C**) Venn diagram illustrating the overlap of FEDGs identified from the two datasets. There are 10 and 15 unique FEDGs in the GSE75214 and GSE112366 datasets, respectively, with 7 FEDGs overlapping between them.

**Figure 5 ijms-27-00019-f005:**
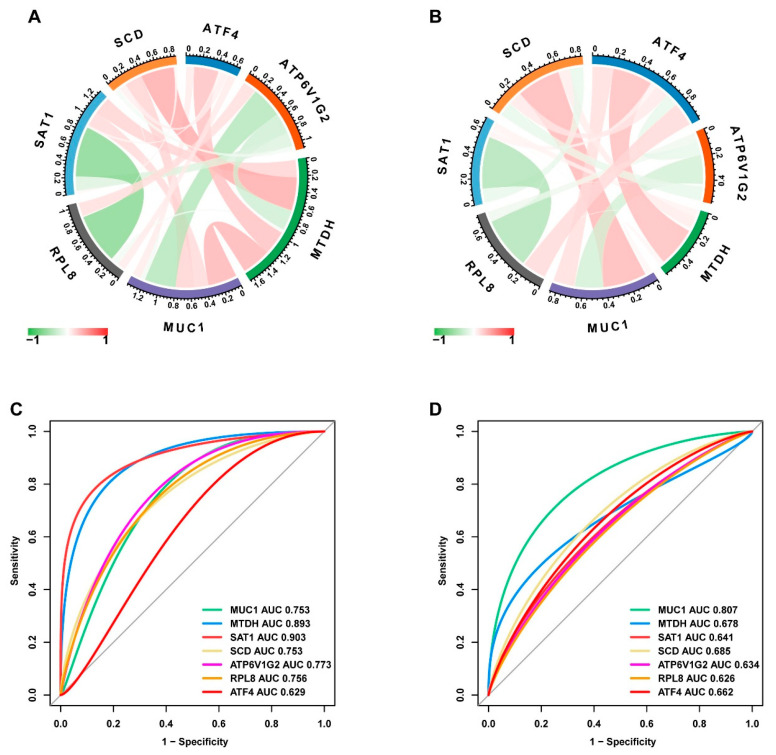
Identification and validation of hub FEDGs. (**A**,**B**) Circular chord plots visualizing the identified hub FEDGs (including *MUC1*, *ATF4*, *SCD*, etc.) and their correlations in the GSE75214 (**A**) and GSE112366 (**B**) datasets, respectively. (**C**,**D**) Receiver operating characteristic (ROC) curves evaluating the diagnostic performance of the individual hub FEDGs in the GSE75214 (**C**) and GSE112366 (**D**) datasets. The area under the curve (AUC) value for each gene is indicated.

**Figure 6 ijms-27-00019-f006:**
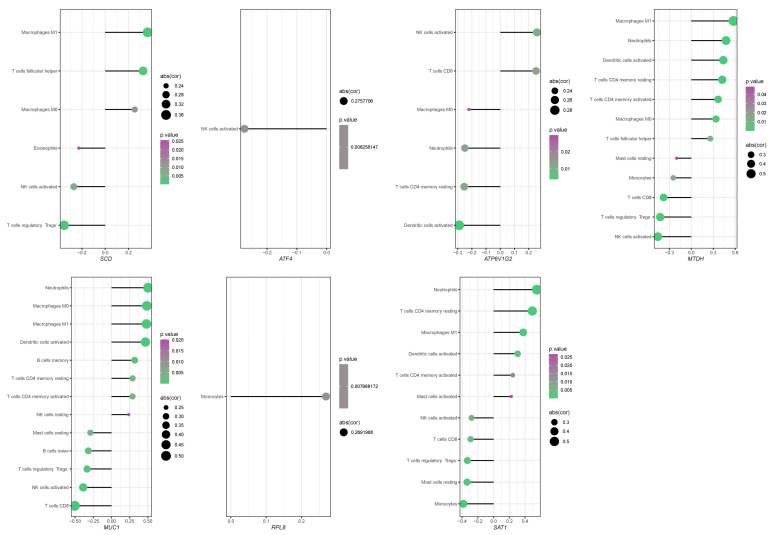
Correlation analysis between the expression of hub FEDG and immune cell infiltration levels. The lollipop diagram illustrates the correlation coefficients of seven key genes (*SCD*, *ATF4*, *ATP6V1G2*, *MTDH*, *MUC1*, *RPL8*, *SAT1*) with the abundance of various infiltrating immune cell types. Only immune cell types with a statistically significant correlation (*p* < 0.05) for each gene are displayed. (Plotting rules: Each dot represents the correlation between a gene and an immune cell type. Dot position on the *X*-axis indicates the Pearson correlation coefficient. Dot size is proportional to the absolute value of cor. Dot color denotes the statistical significance.).

**Figure 7 ijms-27-00019-f007:**
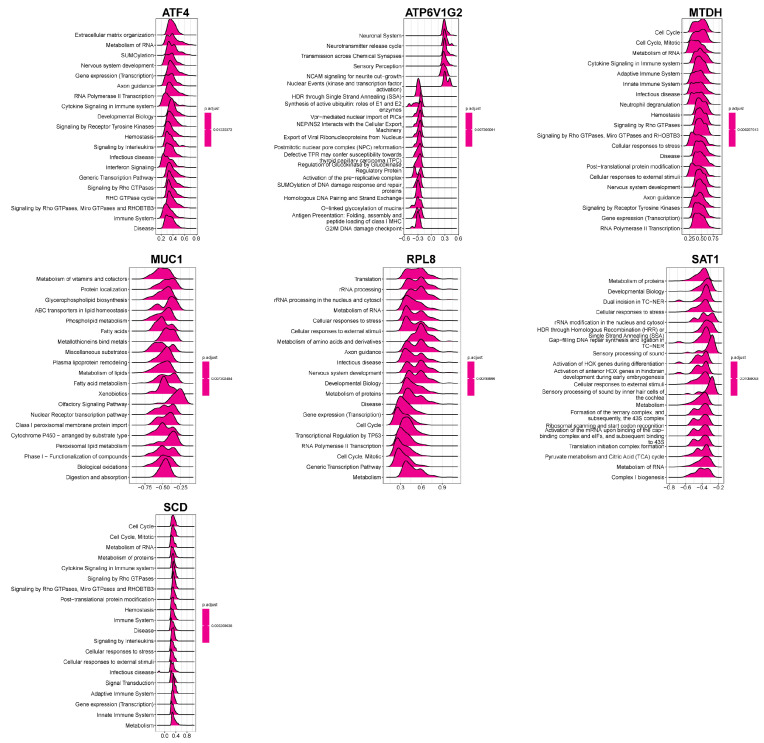
Functional enrichment analysis of hub FEDG in biological processes. Violin plots displaying the enrichment distribution of seven genes (*ATF4*, *ATP6V1G2*, *MTDH*, *MUC1*, *RPL8*, *SAT1*, and *SCD*) across multiple biological processes.

**Figure 8 ijms-27-00019-f008:**
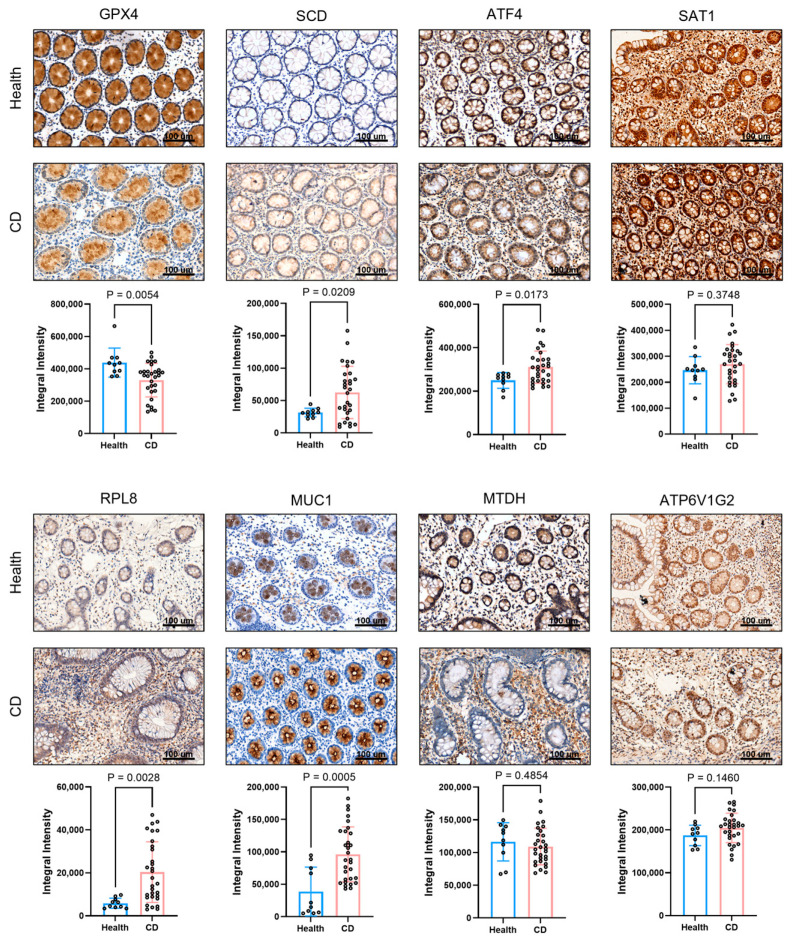
Analysis of immune-related protein expression in intestinal mucosal tissues from healthy individuals and Crohn’s disease (CD) patients. Representative immunohistochemical staining images of eight proteins (GPX4, SCD, ATF4, SAT1, RPL8, MUC1, MTDH, and ATP6V1G2) showing brownish-positive signals. Scale bar: 100 μm and quantitative comparison of the positive area percentage for all eight proteins between healthy and CD groups.

**Figure 9 ijms-27-00019-f009:**
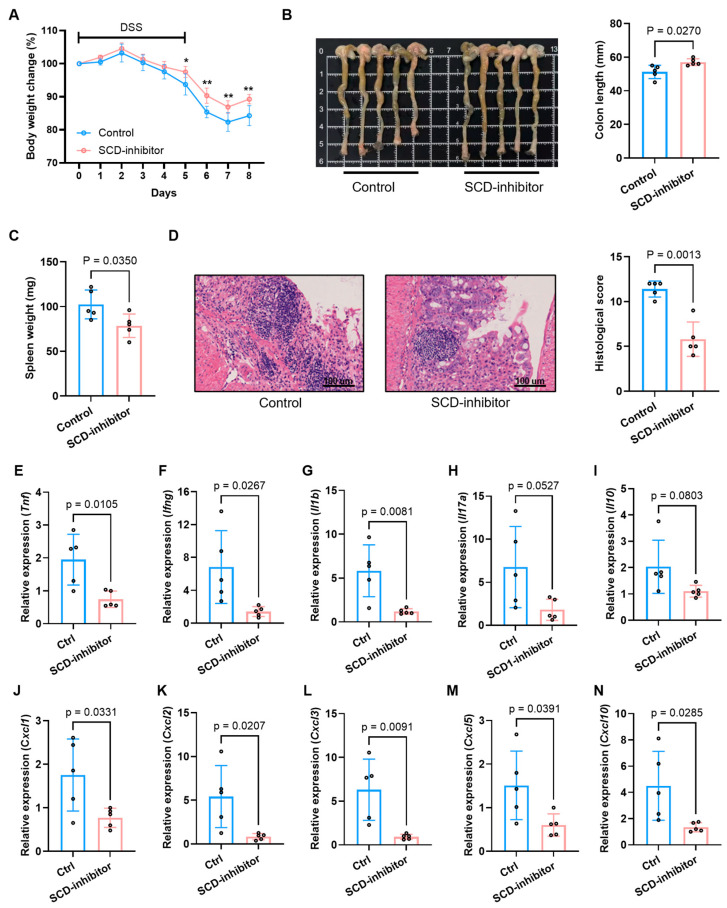
Therapeutic effects of SCD inhibition on DSS-induced colitis in mice. (**A**) Body weight change trends in control and SCD-inhibitor treated groups during colitis induction. (**B**) Macroscopic colon appearance (**left**) and quantitative colon length measurements (**right**) between experimental groups. (**C**) Splenic weight comparison showing reduced splenomegaly after SCD inhibition. (**D**) Representative colon histology (H&E staining) and corresponding histopathological scores demonstrating tissue protection. (**E**–**N**) mRNA expression levels of pro-inflammatory cytokines (**E**–**I**) and chemokines (**J**–**N**) in colonic tissues measured by RT-qPCR. Data are presented as mean ± SD, each dot represents an independent mouse sample (* *p* < 0.05, ** *p* < 0.01).

**Figure 10 ijms-27-00019-f010:**
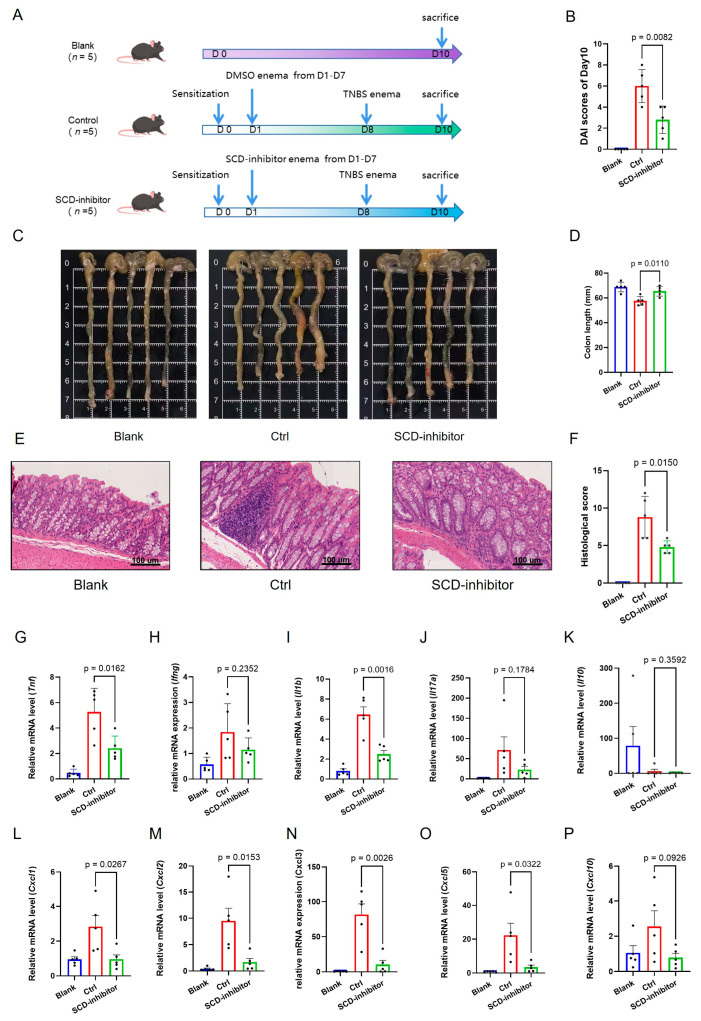
Therapeutic effects of SCD inhibition on TNBS-induced colitis in mice. (**A**) Schematic showing administration schedule for TNBS. (**B**) Disease Activity Index (DAI) scores of Day 10 in the Blank, Control and SCD-inhibitor4 treated groups. (**C**,**D**) Macroscopic colon appearance and quantitative colon length measurements between experimental groups. (**E**,**F**) Representative colon histology (H&E staining) and corresponding histopathological scores demonstrating tissue protection. (**G**–**P**) mRNA expression levels of pro-inflammatory cytokines and chemokines in colonic tissues measured by RT-qPCR. Data are presented as mean ± SD, each dot represents an independent mouse sample.

## Data Availability

GSE112366 and GSE75214 datasets were downloaded from the GEO database (https://www.ncbi.nlm.nih.gov/geo, access date: 1 June 2022). The ferroptosis-related genes (FRGs) were obtained from the FreeDb online database (http://www.datjar.com:40013/bt2104/, access date: 1 June 2022).

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
