# Peer review of "Integrating Bioinformatics and Experimental Validation Identifies SCD as a Ferroptosis-Related Immune Regulator and Therapeutic Target in Crohn’s Disease"

_ijms, 2025, doi:10.3390/ijms27010019_

Round 1

Reviewer 1 Report

Comments and Suggestions for Authors

This manuscript investigates the role of ferroptosis-related genes (FRGs) in the progression of Crohn’s disease (CD), a chronic inflammatory bowel disease. By integrating bioinformatics analyses of transcriptomic datasets with experimental approaches, the authors identified key FRGs associated with inflammation in CD. Special attention is given to stearoyl-CoA desaturase 1 (SCD1), which was found to be implicated in ferroptosis and immunomodulation within the disease context. The study highlights the importance of FRGs in medi pathogenic processes in CD and suggests they may serve as promising targets for therapeutic intervention. Moreover, the work discusses current gaps in the mechanistic understanding of CD and ferroptosis, calling for further research into gene-driven mechanisms underlying intestinal inflammation.However, there are many problems in the current manuscript. Hence, I would like to recommend reject of this manuscript.

1.The disease investigated in this article is Crohn's disease, but DSS-induced colitis is a classic model for ulcerative colitis. The authors should have used the TNBS model, which would have been more appropriate.

2.In Methods  4.1 , the authors state that differentially expressed genes were selected with |log2FC|> 1. However, in Figures 1A and 1C, the genes shown as differentially expressed do not meet the criterion of |log2FC|> 1.

3.In the supplementary Figure 4, compared to the control group, the expression of SCD1 in the mouse colitis model did not change (P=0.6372). Therefore, is it meaningful to intervene in SCD1?

4.There are also many minor errors in the manuscript, such as in line 98,"of which 9a11 were  upregulated and 857 were downregulated (Figures 2A and 2B)."

Author Response

Reply to Reviewer #1

Dear Reviewers,

We thank the reviewer for the thorough and critical assessment, which has been instrumental in improving our work.

Comment 1:

"The disease investigated in this article is Crohn's disease, but DSS-induced colitis is a classic model for ulcerative colitis. The authors should have used the TNBS model, which would have been more appropriate."

Response:
We sincerely thank the reviewer for this crucial suggestion. We have conducted a new set of experiments using the TNBS-induced colitis model in mice.

The results from this new model are highly consistent with our original conclusions:

(a) Local inhibition of SCD in the TNBS model significantly ameliorated disease severity, as evidenced by reduced body weight loss, preserved colon length, and improved histopathological scores.

(b) Similar to the DSS model, SCD inhibition in the TNBS model potently suppressed the expression of key pro-inflammatory cytokines (e.g.,Tnf, Il1b) and chemokines.

The results mentioned have been added in the revised manuscript as Figure 10

Comment 2:

"In Methods 4.1 , the authors state that differentially expressed genes were selected with |log2FC|> 1. However, in Figures 1A and 1C, the genes shown as differentially expressed do not meet the criterion of |log2FC|> 1."

Response: We sincerely thank the reviewer. There was a typographical error in the Methods 4.1 section. The actual threshold used for identifying differentially expressed genes in our analysis was |log2FC| > 0 with a P-value < 0.05. We have now corrected this error in the revised manuscript. We apologize for any confusion this may have caused.

Comment 3:

"In the supplementary Figure 4, compared to the control group, the expression of SCD in the mouse colitis model did not change (P=0.6372). Therefore, is it meaningful to intervene in SCD?."

Response: We thank the reviewer for this important question. Although the protein level of SCD did not show a significant change in the DSS model, our subsequent animal intervention experiments directly demonstrate that inhibiting SCD is meaningful.

Comment 4:

"There are also many minor errors in the manuscript, such as in line 98,"of which 9a11 were upregulated and 857 were downregulated (Figures 2A and 2B)."

Response:

We appreciate the comments. We have revised the manuscript at our best. Those changes or corrections have been highlighted in RED in the revised manuscript.

Reviewer 2 Report

Comments and Suggestions for Authors

The authors have nicely narrowed down genes that could be targeted to improve outcome for Crohn's disease using published datasets and then conducting their own experiments using the patients samples and the DSS model. The identification of SCD as a marker with its inhibition showing promising outcome in mice provides a path for future studies in this direction to see if patients can actually benefit from SCD inhibition. Few suggestions to the authors:

  1. The authors have good immunohistochemical stainings but given that the architecture is somewhat different when comparing healthy and Crohns's disease patients, it might be good to represent quantitation incorporating intensity and area and not just area. Parameters like integral intensity or intensity density could be used for quantitation.
  2. For the immunohistochemical stainings in the DSS treated mice, the stainings are very dark and could make it difficult to quantify if the stainings are in the saturated range.
  3. The authors are using GPX4 as a marker for ferroptosis but that is not sufficient. Did the authors try prussian blue staining to visualize if there were differences in iron levels?
  4. For the in vivo study using the SCD inhibitor, did the authors try IHC to see if there were potential correlations with the other hub genes upon inhibition of SCD?

Author Response

Response to Reviewer #2

We are grateful for the reviewer's positive feedback and valuable suggestions.

Comment 1:

"The authors have good immunohistochemical stainings but given that the architecture is somewhat different when comparing healthy and Crohns's disease patients, it might be good to represent quantitation incorporating intensity and area and not just area. Parameters like integral intensity or intensity density could be used for quantitation.."

Response:
We thank the reviewer for this excellent suggestion. We have now re-analyzed all IHC data (from both human samples and mouse models) using integrated density (IntDen). The revised quantitative analyses for Figure 8 and Supplementary Figure 4 have been updated accordingly in the manuscript.

Comment 2:

"For the immunohistochemical stainings in the DSS treated mice, the stainings are very dark and could make it difficult to quantify if the stainings are in the saturated range."

Response:
We thank the reviewer for this thoughtful observation. To ensure accurate quantification, we have re-captured the microscopic images of the IHC staining to prevent signal saturation. These new, clearer images have been included in the revised Supplementary Figure 4 and are now suitable for reliable quantitative analysis.

Comment 3:

"The authors are using GPX4 as a marker for ferroptosis but that is not sufficient. Did the authors try prussian blue staining to visualize if there were differences in iron levels?"

Response:
We thank the reviewer for this valuable suggestion. In direct response to the reviewer's comment, we performed Prussian blue staining, but we did not observe significant blue staining in colon tissues from either control or DSS-treated mice.We further investigated the reason for this result and found that Prussian blue stain (Beyotime, C0127S) is designed to detect trivalent ferric iron (Fe³⁺). However, the hallmark of ferroptosis is the accumulation of labile divalent ferrous iron (Fe²⁺).

Comment 4:

"For the in vivo study using the SCD inhibitor, did the authors try IHC to see if there were potential correlations with the other hub genes upon inhibition of SCD?"

Response:
This is a very insightful suggestion. We have added IHC for the other two key hub genes, ATP6V1G2 and SAT1, on colon tissues from the DSS and TNBS model, which is briefly discussed in the Results.

Reviewer 3 Report

Comments and Suggestions for Authors

As a key signaling molecule in ferroptosis, SCD1 is investigated in this study through bioinformatic analysis to explore its potential as a therapeutic target for Crohn's disease (CD), with validation via animal experiments. While the manuscript design is appealing, several issues need to be addressed:

  1. Numerous signaling molecules regulate ferroptosis. What are the advantages of focusing on SCD1 compared to others? Could the bioinformatic analysis include comparisons with other critical ferroptosis regulators to highlight the rationale for selecting SCD1?

  1. Ferroptosis, as a significant form of programmed cell death, plays a role in various intestinal diseases. How can the specificity of SCD1 in CD—particularly in comparison to other intestinal disorders such as ulcerative colitis (UC)—be demonstrated?

  1. In the animal experiments, only a comparison between the model mice and those treated with an SCD1 inhibitor was provided. Including a control group of normal mice would more clearly illustrate the effects of the SCD1 inhibitor.

  1. Based on my research experience, the DSS-induced colitis model more closely resembles the disease characteristics of clinical UC patients, whereas the TNBS-induced colitis model is more representative of CD.

  1. Some expressions require careful review and revision. For instance, "SCD1" is frequently abbreviated as "SCD" in later sections. Additionally, in the ethics statement, aside from approval for human studies, the ethical approval number and documentation for the animal experiments should also be provided.

Author Response

Response to Reviewer #3

We appreciate the reviewer's constructive comments, which have helped us better contextualize our work.

Comment 1:

"Numerous signaling molecules regulate ferroptosis. What are the advantages of focusing on SCD compared to others? Could the bioinformatic analysis include comparisons with other critical ferroptosis regulators to highlight the rationale for selecting SCD?"

Response:

We appreciate the comments. We focused on SCD because it consistently emerged as a key regulator across our analyses. It was identified as a hub gene in both independent datasets (GSE75214 and GSE112366) and its protein upregulation was validated in human CD tissues via IHC, demonstrating robust clinical relevance.

Comment 2:

"Ferroptosis, as a significant form of programmed cell death, plays a role in various intestinal diseases. How can the specificity of SCD in CD—particularly in comparison to other intestinal disorders such as ulcerative colitis (UC)—be demonstrated?"

Response:
We thank the reviewer for this insightful comment. The primary goal of this study was to conduct a deep mechanistic investigation into the role of SCD and ferroptosis in Crohn's disease (CD).We agree that SCD may also have functional importance in other intestinal diseases like UC. Elucidating its potential broader roles across different diseases will be a valuable objective for our future research.

Comment 3:

"In the animal experiments, only a comparison between the model mice and those treated with an SCD inhibitor was provided. Including a control group of normal mice would more clearly illustrate the effects of the SCD inhibitor."

Response:
We agree. The new animal experiment figures (TNBS model) now explicitly include data from the normal control group (without colitis induction) to more clearly illustrate the therapeutic effect of the SCD inhibitor.The results mentioned have been added in the revised manuscript and can also be seen as Figure 10.

Comment 4:

"Based on my research experience, the DSS-induced colitis model more closely resembles the disease characteristics of clinical UC patients, whereas the TNBS-induced colitis model is more representative of CD."

Response:
We thank the reviewer for reinforcing this critical point. We have now fully addressed this by including a complete validation in the TNBS-induced colitis model. The concordant results between both models greatly enhance the robustness and disease-relevance of our conclusions.The results mentioned have been added in the revised manuscript and can also be seen as Figure 10. 

Comment 5:

"Some expressions require careful review and revision. For instance, "SCD1" is frequently abbreviated as "SCD" in later sections. Additionally, in the ethics statement, aside from approval for human studies, the ethical approval number and documentation for the animal experiments should also be provided."

Response:
We have thoroughly proofread the manuscript to ensure consistent use of terminology (SCD) . Regarding the animal ethics statement, we have now provided the ethical approval number for the animal experiments in the Institutional Review Board Statement of the revised manuscript. We apologize for this omission in the previous version.

Round 2

Reviewer 3 Report

Comments and Suggestions for Authors

The author's correction has dispelled my doubts and I suggest accepting it.